# Development of a New Indirect ELISA Test for the Detection of Anti-Feline Coronavirus Antibodies in Cats

**DOI:** 10.3390/vetsci12030245

**Published:** 2025-03-04

**Authors:** Irene Ferrero, Sarah Dewilde, Paolo Poletti, Barbara Canepa, Enrica Giachino, Paola Dall’Ara, Joel Filipe

**Affiliations:** 1Agrolabo S.p.A., 10010 Scarmagno, Italy; 2Gem Forlab S.r.l., 10014 Caluso, Italy; 3Department of Veterinary Medicine and Animal Sciences, University of Milan, 26900 Lodi, Italy

**Keywords:** FCoV, FECV, FIPV, Coronavirus, FIP, FCoVCHECK, ELISA, IFAT

## Abstract

The aim of this work was the development of a new rapid indirect ELISA assay, called FCoVCHECK Ab ELISA, for the detection of antibodies against the feline coronavirus (FCoV) in feline serum/plasma samples. The more virulent form of FCoV may cause feline infectious peritonitis (FIP), a progressive and often fatal disease. The diagnosis of FIP is complex, and serological examinations, such as IFAT, considered the gold standard, and indirect ELISA are widely used. The new assay is the result of an extensive set-up phase that considered different concentrations of reagents and several methods to assess the cut-off to achieve the best test performance in terms of sensitivity, specificity and accuracy. Compared to IFAT as a reference, the developed ELISA agreed at 96.4% (93.5% sensitivity, 95% confidence interval (CI): 83.5–97.9%; 100% specificity, 95% CI: 90.8–100%). In addition, the new ELISA showed the best performance compared with two widely used commercial ELISA assays and it provided correct and reliable results quickly (1 h). It does not require thermostats, contains safe reagents for the end-user and has long-term storage, up to 18 months at +2–8 °C. All these features make this test optimal for use in veterinary clinics and practices.

## 1. Introduction

Feline coronavirus (FCoV) is an enveloped single-stranded RNA virus that belongs to the family Coronaviridae. The genome of the virus encodes four major structural proteins, such as the spike (S), nucleocapsid (N), envelope (E) and membrane (M) proteins, and seven nonstructural proteins, including two replicase proteins (1a, 1b) and five accessory proteins (3a, 3b, 3c, 7a and 7b). FCoV is divided into two serotypes according to its antigenicity: FCoV I, the most diffuse among cats in terms of natural infections worldwide, and FCoV II, less common, originating from recombination between FCoV I and canine coronavirus (CCoV). Both types of FCoV can be divided into two biotypes based on pathogenicity: feline enteric coronavirus (FECV) and feline infectious peritonitis virus (FIPV) [1,2]. FECV infections are usually benign, and only rarely can they induce severe enteritis. Instead, the FIPV biotype is more virulent and causes feline infectious peritonitis (FIP), a progressive and often fatal disease. The widely accepted hypothesis is that FECV converts to FIPV, so genetic variation and subsequent selection facilitate the switching of cell tropism from enterocytes to systemic monocytes/macrophages within an FCoV-infected cat. Effective replication of the virus within this type of leukocytes is necessary for the development of FIP [3,4]. More recently, two amino acid substitutions (M1058L and S1060A) within the spike S protein have been analyzed and linked to the systemic spread of FCoV from the intestine rather than a virulence change, as previously assumed [5].

In general, FECV proliferation is confined to intestinal mucosal epithelial cells or mesenteric lymph nodes, causing mild and generally self-limiting infections [6], whereas FIPV infects feline peritoneal macrophages and other types of cells, including monocytes, plasma cells, lymphocytes and neurocytes, resulting in high titers of the virus and systemic and fatal disease [7]. FIP arises only in a small percentage of FCoV-infected cats following FCoV infection, and its occurrence is closely related to crowded and unsanitary environments, virus virulence and host genetics factors, age of the cat, concurrent diseases or other stressors [1,3,8].

The clinical symptoms of FIP vary considerably, reflecting the variability in the distribution of vasculitis and granulomatous lesions. The vasculopathy can result in thoracic, abdominal and pericardial effusions, thus resulting in the so-called “wet” or “effusive” FIP, while granuloma formation, mainly in the abdominal organs, results in the “dry” or “non-effusive” FIP. However, there is a considerable overlap between the two forms [1]. These forms depend on the strength of the T cell-mediated response. In fact, progression of the infection to FIP may be the consequence of severe immunodepression by T cell depletion [9]. The wet forms are presumed to arise as a consequence of weak cell-mediated immune responses. The clinical aspect of FIP is highly variable. Fever that is refractory to antibiotics, lethargy, anorexia and weight loss are common non-specific signs. Ascites is the most obvious manifestation, along with thoracic and pericardial effusion of the effusive form [9].

The diagnosis of FIP is complex and requires various investigations since existing diagnostic tests cannot differentiate between FECV and FIPV [10]. The clinician should consider the history and symptoms, along with the results of histological, biochemical and serological examinations of the individual patient. Measurement of anti-FCoV antibodies is the most widely used method to detect FCoV infection or exposure. In fact, antibodies can be present in absence of acute infection due to previous contact with the pathogen and cleared infection. However, although cats with FIP tend to have higher FCoV antibody titers than cats without FIP [8,9], the presence of antibodies is not useful for diagnosing FIP since antibodies are not only present in cats with FIP but also in healthy FCoV-infected cats or FCoV-infected cats with other diseases [8,9].

Tests for FCoV antibodies detection include indirect immunofluorescent antibody tests (IFAT), considered the gold standard [11,12,13], using feline kidney cells infected with FCoV, or the related porcine transmissible gastroenteritis virus (TGEV), ELISA assays and rapid immunochromatographic (RIM) tests [1,10]. Another method, immunoblotting, is only available commercially in specialized laboratories [12].

The aim of this work was the development of a new simple and rapid indirect assay, called FCoVCHECK Ab ELISA, for the detection of FCoV antibodies in feline samples, which is easy to execute and to read and able to provide results comparable to that would be obtained with the reference IFAT assay.

## 2. Materials and Methods

### 2.1. Samples Collection

The animal samples (serum/plasma) used in this study were humanely collected under informed consent by clinicians during normal routine analysis, in agreement with animals’ owners, and donated for the successful completion of the project. Each sample was divided into small aliquots (5–10 µL) upon arrival at the laboratories and stored at −70 °C. Once thawed, the aliquots were used for the analytical sessions, then the leftovers were discarded. The procedures performed are not considered animal experimentation and therefore are not under the Italian Legislative Decree 26/2014 and Directive 2019/63/EU of the European Parliament.

In order to set up the new ELISA and assess its reliability, a total of 110 sera/plasma samples from cats with various clinical conditions were collected from 4 veterinary clinics in northern and central Italy. All the samples used in this study were first tested for FCoV antibodies by IFAT kit Fluo Feline Coronavirus (Agrolabo S.p.A., Scarmagno, Italy) and thus classified as positives or negatives. Some of these samples were highly lipemic or hemolytic. No information was provided about breed, sex, age, clinical outcomes, disease progression or treatments of patients with drugs; thus, no sample selection was performed to contemplate the broadest possible cases.

### 2.2. ELISA Plate Preparation

Nunc MaxiSorp and Nunc PolySorp microplates (Thermo Fisher Scientific, Waltham, MA, USA) were used in the setup of the test.

All the coating and blocking conditions used in the development of the new indirect ELISA test are described in standardized protocols [14,15,16] and based on internal company protocols and experience. In the preliminary assays, microplates were coated with an antigen derived from FCoV Type I (WSU 79-1146 strain, LGC Limited, Teddington, UK) at different concentrations, ranging from 1 μg/mL to 20 μg/mL, in coating buffer formed by carbonate–bicarbonate (CB) buffer, with a pH of 9.6, and incubated at +2–8 °C overnight (O/N). After a washing step with wash buffer (Agrolabo S.p.A., Scarmagno, Italy), the plates were successfully blocked with two types of blocking buffers, blocking buffer 1, a BSA-free buffer containing MOPS (3-(N-morpholino)propanesulfonic acid) in a pH range of 6.5–7.5 (Surmodics, Eden Prairie, MN, USA), and blocking buffer 2, formed by phosphate-buffered saline (PBS), added with 1% of bovine serum albumin (BSA) (Agrolabo S.p.A., Scarmagno, Italy). Both were incubated for 1 h at room temperature (RT). The plates were then drained, left to dry and stored at +2–8 °C.

### 2.3. FCoVCHECK Ab ELISA

The new test is the result of an extensive set-up phase carried out by the research and development (R&D) teams of Agrolabo S.p.A. and Gem Forlab S.r.l, considering different concentrations of the antigen, conjugate, controls and sample dilutions, to achieve the best possible test performance in terms of sensitivity, specificity, accuracy, reliability and repeatability of the results.

Nunc PolySorp microplates were prepared as previously described by coating 1 μg/mL of FCoV antigen. In FCoVCHECK Ab ELISA, the positive control (PC), consisting of the unconjugated anti-FIPV antibody (MyBioSource, San Diego, CA, USA) and protein G labeled with horseradish peroxidase (HRP) (Abcam Limited, Cambridge, UK), the negative control (NC) and samples, diluted 1:100 in sample diluent formed by PBS and BSA (Agrolabo S.p.A., Scarmagno, Italy), were distributed 100 µL in each well and incubated 30 min at RT. After a washing step with wash buffer in order to eliminate all the unbound material, 100 µL of the anti-Cat IgG conjugate antibody labeled with HRP (Abcam Limited, Cambridge, UK) were added and incubated for 15 min at RT. After a second round of washes, 100 µL of the substrate/chromogen 3,3′,5,5′-Tetramethylbenzidine (TMB) (Surmodics, MN, USA) were added and incubated for 8 min at RT in the dark, developing a colorimetric reaction (a blue color was indicative of positive samples that contained anti-FCoV antibodies; no color indicated negative ones). Then, the reaction was stopped by adding 100 µL of stop solution, formed by sulphuric acid 0.2 M (Agrolabo S.p.A., Scarmagno, Italy), and the blue reaction turned into shades of yellow. The interpretation of the results was performed by reading the optical density (OD) values at 450 nm using an ELISA microplate reader (Multiskan SkyHigh, Thermo Fisher Scientific, Waltham, MA, USA). The test was considered valid if the positive control had an OD value above 0.6 and the negative control below 0.1. The cut-off was evaluated as described later (Section 2.8), and, finally, the positivity or negativity of each sample was evaluated by calculating the “negative cut-off” (1) and “positive cut-off” (2) based on the OD value of the negative control. The sample was considered negative if the OD value was lower than the negative cut-off, positive if the OD value was greater than the positive cut-off and doubtful if the OD value was between the two cut-offs.Negative cut-off = OD negative control + 0.16(1)Positive cut-off = OD negative control + 0.26(2)

### 2.4. FCoVCHECK Ab ELISA Validation

The FCoVCHECK Ab ELISA was validated by testing 110 feline samples (62 positives and 48 negatives) against the IFAT kit Fluo Feline Coronavirus (Agrolabo S.p.A.), considered as a reference [12], in which the slides contained feline kidney cells infected with TGEV.

The number of samples tested was in line with that used in other bibliographic studies for validation of ELISA tests [17,18,19,20]. Briefly, IFAT was performed by incubating 20 µL of samples diluted 1:100 and controls on substrate slides in a humid chamber for 30 min at +37 °C. After a washing step, one drop (20 µL) of conjugate was added to the wells, and the slides were newly incubated in a humid chamber for 30 min at +37 °C in the dark. After washing, two drops of mounting fluid were added to the slides and the coverslips and then observed under a fluorescence microscope with a filter system for FITC (Axioskop 40, Zeiss, Jena, Germany).

An evaluation was made comparing each sample to the visual intensity of the positive and negative controls’ patterns. A negative test result was given when there was no clear fluorescence or only a weak red-grayish color of the cells. Instead, in positive samples, the cytoplasm and the membrane of the infected cells showed a bright, sharp and clear apple-green fluorescence.

### 2.5. Comparative Study

The new test was compared with two ELISA tests, the INgezim Corona Felino ELISA (Gold Standard Diagnostics, Madrid, Spain) and the ELISA on solid phase (dot assay) ImmunoComb Feline Coronavirus (FCoV) [FIP] Antibody Test Kit (Biogal, Kibbutz Galed, Israel) by analyzing the same 110 feline samples already used in the validation phase, according to the manufacturer’s protocol.

### 2.6. Reproducibility

The ELISA’s overall precision was evaluated by performing tests at different times on the same day and on different days under similar conditions, as mentioned in the guidelines [21]. In our case, tests were carried out by two operators to consider operator-related variables in different analytical sessions, twice a day (morning and afternoon, to consider any changes in ambient temperature, although the ambient temperature in the laboratory was always kept between +20–25 °C) for 6 or 14 consecutive days. The samples were repeated in duplicate for a total of 24 or 56 repetitions for each sample. At the end of the study, test overall precision was defined through the evaluation of variation between wells within a single run of a plate (intra-assay) or between runs (inter-assay) for each sample by evaluating the mean OD, standard deviation (SD) and minimum (Min) and maximum (Max) of coefficient of variation (%CV).

### 2.7. Stability Studies

A single batch of ELISA plates was produced for stability tests. Along with the plates, all the reagents were prepared and then divided for the two types of stability studies to avoid the possibility of mistaken results related to different batches produced. In the real-time stability study, the ELISA kit was stored at +2–8 °C; in the accelerated one, the kit was stored at +37 °C to evaluate the aging process of the product by exposing it to elevated temperatures for a shorter time.

In the absence of specific regulation for stability testing on kits for veterinary use, the Standard Guide for Accelerated Aging of Sterile Medical Device Packages ASTM F1980-21 [22] was used, which applies the Arrhenius equation to estimate the degradation of the medical device under accelerated conditions. In this way, the “Accelerated Aging Time” (AAT), represented by the number of days during which the product is to be tested at the temperature in accelerated condition, was calculated through Formula (3):Accelerated Aging Time (AAT) = Desired Real Time (RT)/Q_10_ ^(T^_AA_^−T^_RT/_^10)^(3)
where “RT” is the desired time at which the product is to be stored, the accelerated aging temperature (T_AA_) is the test temperature in accelerated condition, the real-time temperature (T_RT_) is the temperature at which the product is to be stored and the aging factor (Q_10_) is a measure of how quickly a material system changes when the temperature is increased by +10 °C. It is typically between 1.8 and 2.5, with a value of 2.0 being the most common value. In our case, for the test that would be stored at +2–8 °C for 18 months, the time for accelerated stability study was calculated through the use of calculators found on the web [23,24], and it corresponded to 6 weeks at +37 °C. Therefore, it can be concluded that 1 week at +37 °C is equivalent to 3 months at +2–8 °C. In both the real-time and accelerated stabilities, the first test was performed at the beginning of the study, at time zero (T_0_). The assays were repeated weekly for the first 6 weeks (T_1_–T_6_), with the testing controls and samples with reagents stored at +2–8 °C and at +37 °C. Then, only the plates at +2–8 °C were tested for real-time stability every 3 months up to 18 months (T_7_–T_12_). In each analytical session and for each sample, the percentage remaining activity (%RA) or recovery was calculated and is expressed as the ratio of the obtained OD value to that of time T_0_; the lower limit of acceptability was imposed at 70% [25]. The number of samples analyzed was in line with those used in other stability studies [18,20].

### 2.8. Cut-Off Determination

The ELISA cut-off was determined by comparison of several methods. Initially, it was calculated as the mean OD value of negative samples multiplied by 3 or 4, as described in reference [26], then confirmed through other methods, as well. In one of these, the cut-off was calculated as the mean OD value of negative samples plus 3 times the standard deviation (SD) [26,27,28], considering positive or negative samples with the OD above or below the cut-off, respectively. Samples with OD values within 10% of the cut-off value were considered “doubtful”, as reported in other commercial ELISA assays [29,30]. Moreover, another common method was based on the evaluation of sensitivity and specificity parameters, according to which the best cut-off would be the OD value where the difference between them is smallest [31]. In addition, based on a study present in the literature [26], the optimal cut-off may also be a value between the maximum OD value of negative samples and the minimum OD value of positive samples. Furthermore, the receiver operating characteristics (ROC) curve was evaluated to measure the accuracy of the diagnostic test and to identify the optimal cut-off value that maximizes the difference between true positives and false positives. The ROC curve was constructed by joining the points obtained from the proportion of true positives (sensitivity) and false positives (1-specificity) at all possible cut-off values. The point closest to the upper left corner of the ROC curve usually corresponds to the best cut-off. Closely related to the ROC curve method is the Youden’s index (J), which was calculated for each individual cut-off by Formula (4):J = sensitivity + specificity − 1(4)

The parameter J takes values in the closed range [0, 1]: a value of 0 corresponds to a completely ineffective test, while a value of 1 corresponds to a perfectly effective test. The best cut-off corresponds to the maximum value among the calculated indexes.

In addition, considering the amount of true positives (a), false positives (b), false negatives (c) and true negatives (d) obtained from the FCoVCHECK Ab ELISA against the reference assay, all the diagnostic parameters, including the accuracy, sensitivity, specificity, positive and negative predictive values (PPV and NPV) and positive and negative likelihood ratios (LR+ and LR−) were determined at the selected cut-off value (Formulas (5)–(11)). On the sensitivity, specificity, PPV and NPV parameters, the 95% confidence interval (CI) was also calculated.Diagnostic accuracy = (a + d)/(a + b + c + d)(5)Sensitivity = a/(a + c)(6)Specificity = d/(b + d)(7)Positive predictive value (PPV) = a/(a + b)(8)Negative predictive values (NPV) = d/(c + d)(9)Positive likelihood ratio (LR+) = Sensitivity/(1 − Specificity)(10)Negative likelihood ratio (LR−) = (1 − Sensitivity)/Specificity(11)

### 2.9. Statistical Analysis

Statistical analysis on the results obtained from the comparative study was performed by *t*-test, considering a *p*-value < 0.05 to be significant and by calculating Cohen’s d index (12) to measure the strength of the relationship (effect size). In the formula, M1 and M2 are the mean OD values of the two datasets, and S1 and S2 are their standard deviations.Cohen’s d index = M1 − M2/√ (S1^2^ + S2^2^)/2(12)

In addition, for all the comparison assays, the degree of agreement was calculated by Cohen’s kappa index (13), where P_0_ is the probability of agreement observed, and P_e_ is that obtained by chance.Cohen’s kappa index = P_0_ − P_e_/1 − P_e_(13)

## 3. Results

### 3.1. ELISA General Assay Set Up

Nunc PolySorp microplates were coated with different concentrations of FCoV antigen, from 0.5 µg/mL to 2 µg/mL, in CB buffer, with a pH of 9.6, and blocked with blocking buffer 1. The ELISA preliminary test was performed with one positive and one negative sample diluted 1:200, 50 µL volume added to each well and incubated for 30 min at RT. The conjugate antibody was tested at dilutions 1:2500–1:10,000 and incubated for 15 min at RT, while the TMB substrate was incubated for 8 min. We achieved good performances in terms of cost and efficiency with 1 µg/mL of antigen and conjugate antibody 1:2500, while the other conditions gave only a weak positive signal. As expected, the OD values were higher in the plates coated with 2 µg/mL of antigen and better with conjugate antibody 1:2500 or 1:5000. Regarding the negative sample, we obtained good results with low OD values (<0.1), also at higher concentrations of the conjugate antibody (Table 1).

Preliminary ELISA assays were also performed on Nunc MaxiSorp microplates, but too much antigen would be needed in terms of performance and cost. In that case, the plates were coated with a concentration ranging from 0 µg/mL to 20 µg/mL of antigen, diluted in two types of coating buffers, the CB buffer pH 9.6 or PBS, then blocked with blocking buffer 2. ELISA assays were performed with the positive and negative samples diluted 1:200 (50 µL/well) and the conjugate antibody at dilutions 1:5000–1:20,000. The samples were incubated for 30 min, the conjugate antibody for 15 min and TMB for 8 min at RT. We obtained good performances with both coating buffers despite the OD values of the positive sample being higher and thus better for a qualitative assay when coating was performed in CB buffer than in PBS. The best results were obtained with 10 or 20 µg/mL of antigen diluted in CB buffer pH 9.6 and conjugate 1:5000 (Appendix A). From these data, we selected Nunc PolySorp plates for subsequent tests because the performances were better than on MaxiSorp ones in terms of cost and efficiency (a lesser amount of antigen was required for the coating plates).

### 3.2. Sample Dilution Selection

To further improve the signal on the plates coated with 1 µg/mL of antigen, we first performed an ELISA assay by increasing the sample volume to be analyzed from 50 µL to 100 µL, keeping the dilution 1:200 and the conjugate antibody at 1:2500. The FCoVCHECK Ab ELISA correctly identified all the positive and negative samples (Table 2).

Then, to evaluate the best test performance, two sample dilutions were tested. For this purpose, 110 feline samples, including 62 positives and 48 negatives in IFAT, considered the reference test, were diluted 1:100 or 1:200, and the conjugate antibody was used at dilution 1:3000, slightly higher than in the previous test. The incubation times of the samples, conjugate and TMB substrate were the same as in the previous ELISA assays. The cut-off was estimated in a general and simple manner, as described in references [26,27,28], as the mean OD value of the negative samples plus 3 times the SD, thus considering positive or negative samples with OD values above or below the cut-off, respectively. The detailed data are reported in Appendix A. In the first ELISA assay on samples diluted 1:100, the cut-off value was set at OD 0.303. The FCoVCHECK Ab ELISA identified 58 positive samples out of 62, 48 negatives out of 48 and 4 false negatives (Figure 1a). Test sensitivity and specificity were 93.5% and 100%, respectively (Table 3). When the samples were diluted 1:200, the cut-off value was set at OD 0.260: the FCoVCHECK Ab ELISA identified 47 positive samples out of 62, 48 negatives out of 48 and 15 false negatives (Figure 1b). The test sensitivity and specificity were 75.8% and 100%, respectively (Table 3). Since sample dilution 1:100 improved the test sensitivity, we decided to use it and the conjugate antibody diluted 1:3000 in the ELISA validation phase and future assays.

### 3.3. Positive and Negative Control Evaluation

Initially, the unconjugated goat anti-FIPV antibody to be used as a positive control was evaluated for aspecific binding with the plate components. For this purpose, only the anti-FIPV antibody was added to the plate at dilutions ranging from 1:50 to 1:10,000 and detected by two types of conjugate antibodies, the anti-Cat IgG and the anti-Goat, both labeled with HRP. A strong positive control should have an OD ≥ 1.5, with a minimum OD value of 0.6 for test validity, as reported in references [17,32]. In fact, in an ELISA assay that uses TMB as substrate, a weak positive signal has an OD value of about 0.3 at 450 nm.

As expected, the signals derived from the anti-Cat IgG antibody resulted in negative OD values (OD < 0.1), indicative of the absence of aspecific binding, compared with those obtained with the anti-Goat one (OD > 1.0). In detail, the dilutions from 1:50 to 1:5000 indicated high-affinity binding to the coated antigen (Table 4).

Since the anti-FIPV antibody was unconjugated, we developed a positive control independent from the conjugate binding formed by a mixture of the anti-FIPV antibody and protein G labeled with HRP. Protein G is known to bind the constant portions of immunoglobulins [33]; thus, in our case, we assumed that protein G could have bound to the anti-FIPV antibody, and we performed several assays to determine the feasibility of this assumption. To identify which combination of antibody and protein G-HRP was optimal for a good positive control, various concentrations of the two reagents were tested to obtain an OD value at least above 0.6 (Appendix A). We found four good combinations, but finally, we chose the mixture formed by the anti-FIPV antibody diluted 1:600 and protein G-HRP diluted 1:800 for cost and performance reasons. In fact, when the anti-FIPV antibody was diluted 1:600, the better concentration of protein G-HRP that gave a higher OD value was 1:800 (Figure 2). Regarding the negative control, we tested the sample diluent 24 times. The mean OD value was 0.043 and did not exceed the OD limit of 0.3 [17,32]. Moreover, as a comparison, we considered the results obtained by testing the negative samples (diluted 1:100, used previously) to verify if they gave a signal similar to that obtained with the negative control. From the data analysis, we concluded that the mean OD value of the negative control formed by sample diluent was similar to that obtained from the negative samples (Appendix A), thus making its use feasible.

### 3.4. ELISA Validation

The FCoVCHECK Ab ELISA was validated against the reference test IFAT by using the same 110 samples used previously (62 positives and 48 negatives) diluted 1:100. Of 110 samples tested, the FCoVCHECK Ab ELISA detected 58 positives out of 62 and 52 negatives out of 48. In Figure 3 are shown some examples of the IFAT results. The cut-off value, calculated by the mean OD value of the negative samples plus 3 times the SD [26,27,28], was set at OD 0.303. Considering a 10% uncertainty level around the cut-off value as a “doubtful zone” (OD values between 0.273 and 0.334), four samples resulted in false negatives in ELISA (n. 24, 29, 67 and 76, with OD values between 0.070 and 0.110), while one (n. 53) resulted in “doubtful”, with an OD value of 0.289 (Table 3, Appendix A). Compared to IFAT, the FCoVCHECK Ab ELISA agreed at 96.4%, with 93.5% of sensitivity (95% CI: 83.5–97.9%) and 100% of specificity (95% CI: 90.8–100%). Cohen’s kappa was 0.925, indicative of very good agreement. The PPV was 100% (95% CI: 92.3–100%), and the NPV was 92.3% (95% CI: 80.6–97.5%).

### 3.5. Determination of Cut-Off Value

The cut-off of FCoVCHECK Ab ELISA was determined on the 110 feline samples already tested in the validation phase by taking into account IFAT as the reference test, and it was initially set at OD 0.303 by calculating the mean OD value of the negative samples plus 3 times the SD [26,27,28] and by considering a 10% uncertainty level around the cut-off value [29,30]. Then, these results were confirmed by comparison with other methods. At first, the best cut-off corresponded to an OD value of 0.250, at which the difference between the sensitivity and specificity parameters was minimal (difference equal to 0.002), as reported in reference [31]. At this cut-off value, the sensitivity and specificity were 93.5% and 93.8%, respectively (Figure 4a; Appendix A). According to another study present in the literature [26], a correct cut-off may be a value between the maximum OD value of the negative samples and the minimum OD value of the positive ones corresponding to OD values of 0.289 and 0.304, respectively. Based on the ROC curve, the best cut-off corresponded to an OD value of 0.290 (Figure 4b) and, according to the Youden’s index, the maximum calculated J value was 0.935, which coincides with OD cut-off values of 0.290 and 0.300 (Figure 4c). In conclusion, the exact cut-off of the FCoVCHECK Ab ELISA corresponded to OD 0.290. However, since all the sensitivity and specificity parameters remained the same until OD 0.300 (Appendix A), this OD value was considered as the cut-off in the final kit, confirming the results obtained in the validation phase.

### 3.6. Reproducibility Study

A reproducibility study was performed by testing the PC, NC, three negative samples and one positive sample in duplicate, twice in a day, for 6 or 14 consecutive days (24 or 56 tests for each sample) (Appendix A). In the ELISA, the intra-assay %CV should be lower than 10% and the inter-assay %CV lower than 15–20% limits, respectively. However, 20% is defined as the limit of acceptable %CV for both types of assays when considering precision and accuracy [25,34]. The FCoVCHECK Ab ELISA was accurate in both the intra-assay and inter-assay, with the %CV results within the limits (Table 5, Appendix A).

### 3.7. Comparison Analysis

The 110 samples already tested in the validation phase were also analyzed by two other ELISA tests, the ImmunoComb Feline Coronavirus Antibody Test Kit (Biogal) and the INgezim Corona Felino ELISA (Gold Standard Diagnostics). In all cases, IFAT was considered as an additional test of comparison (Appendix A).

#### 3.7.1. Comparative Study with ImmunoComb Feline Coronavirus Antibody Test Kit (Biogal)

The FCoVCHECK Ab ELISA agreed with the ImmunoComb test kit at 93.6% (Cohen’s kappa: 0.872, very good agreement). Of 110 samples tested, the ImmunoComb assay detected 45 negatives and 65 positives, including seven false-positive samples (n. 15, 23, 33, 39, 46, 54, 63) (Table 6). Among these seven samples, six were negative and only one (n. 46) was positive in IFAT. Instead, six of these seven samples were positive, and one was negative (n. 15) with the INgezim Corona Felino ELISA. Therefore, we concluded that sample n. 46 was probably a false negative in FCoVCHECK Ab ELISA, and sample n.15 was a false positive in ImmunoComb (Figure 5). In addition, 19 samples resulted in negatives in FCoVCHECK Ab ELISA, but had weak positives (borderline) with a score S2 in ImmunoComb. Among these, only one sample (n. 37) was positive in IFAT. Thus, in addition to sample n. 46 identified previously, n. 37, although with a weak positive in ImmunoComb, could also be a false negative in FCoVCHECK Ab ELISA.

#### 3.7.2. Comparative Study with INgezim Corona Felino ELISA (Gold Standard Diagnostics)

The new test agreed with INgezim Corona Felino ELISA at 82.7% (Cohen’s kappa: 0.646, good agreement). Of 110 samples tested, the INgezim Corona Felino ELISA detected 33 negatives, 58 positives, 2 false negatives and 19 false positives (Table 6). Considering IFAT as a reference, 17 of these 19 samples were negative, and two samples found previously (n. 37 and n. 46) were positive, thus confirming they were false negatives in FCoVCHECK Ab ELISA. When analyzed with ImmunoComb, only 3 of these 17 samples were negative with scores of S0 and S1 (n. 5, 21 and 30), nine samples were weak positive (score S2) and five samples were positive (score ≥ S3). A statistical analysis performed on the mean absorbance values between FCoVCHECK Ab kit and INgezim Corona Felino ELISA found a statistically significant difference only for the negative samples: the mean OD value of the negative samples in the FCoVCHECK Ab ELISA was statistically significantly lower than in the INgezim Corona Felino ELISA. However, the effect size calculated by Cohen’s d gave a value of 0.437, indicative of a small difference (Table 7).

Since cats with FIP tend to have higher FCoV antibody titers than cats without FIP, we also performed a titration comparative assay both with INgezim Corona Felino ELISA and FCoVCHECK Ab ELISA on 21 positive samples, and good correlation and very similar results were obtained in 17 samples. We found small differences of dilution in four samples. In all the cases, very positive samples with very high antibody titers above 1:3200 suggestive of FIP were identified equally by both tests (Appendix A).

#### 3.7.3. Comparison Results of All Tests with IFAT

Considering IFAT as a reference test, of four samples that resulted in false negatives in FCoVCHECK Ab ELISA, two (n. 37, n. 46) were positives in IFAT, ImmunoComb and INgezim Corona Felino ELISA, while the other two samples (n. 98, n. 103) were positive only in IFAT but were negative when analyzed with ImmunoComb and INgezim Corona Felino ELISA. However, since IFAT was used as a reference test, these last two samples were considered false negatives in all the assays. In addition, taking IFAT as a reference, the ImmunoComb kit identified 60 positives out of 62, 42 negatives out of 48, six false positives and two false negatives. The test sensitivity was 96.8% (CI 95%: 87.8–99.4%), and the specificity was 87.5% (CI 95%: 74.1–94.8%). Cohen’s kappa was 0.851, indicative of very good agreement. On the other hand, the INgezim Corona Felino ELISA identified 60 positives out of 62, 31 negatives out of 48, 17 false positives and two false negatives. Test sensitivity was 96.7% (CI 95%: 87.8–99.4%), and the specificity was 64.6% (CI 95%: 49.4–77.4%). Cohen’s kappa was 0.636, indicative of good agreement (Table 8). In conclusion, the FCoVCHECK Ab ELISA had the highest agreement with IFAT and more specificity than the other assays. Moreover, the FCoVCHECK Ab ELISA agreed more with the ImmunoComb assay than the INgezim Corona Felino ELISA.

### 3.8. Stability Study

#### 3.8.1. Accelerated Stability Study

Detailed data, including spectrophotometric readings and percentage remaining activities (%RA), are shown in Appendix A. The PC, NC, three negative samples and two positive samples were tested in the study. In all the tested time points, the remaining activities were above 70% (Table 9). The tests proved that all the reagents in the kit may be stable for 6 weeks at +37 °C, data confirmed by the real-time study that lasted 18 months at +2–8 °C.

#### 3.8.2. Real-Time Stability Study

The real-time stability study was conducted for 18 months, during which the kits were stored at +2–8 °C. It was carried out with the same controls used in the accelerated stability assays but with other samples because of their insufficient quantity for all the assays. In particular, we tested two negative and three positive samples. All the tests confirmed the results obtained with the accelerated stability, as the %RA values remained high, above 70% (Appendix A) and proved that all the reagents in the kit may be stable for 18 months at the storage temperature +2–8 °C.

## 4. Discussion

This work contributes to the development of a new ELISA test for anti-coronavirus antibody detection in feline serum or plasma samples. Nunc MaxiSorp and Nunc PolySorp microplates were selected based on their wide use in the literature and the company’s extensive in-house ELISA testing experience. Both types of plates are commonly used in ELISA assays to immobilize antigens or antibodies. Nunc PolySorp plates have a hydrophobic surface and thus permit the adsorption of hydrophobic molecules. They are used to immobilize both antigens [35,36] and antibodies [37] in ELISA assays. Nunc MaxiSorp microplates have a hydrophilic surface treatment optimized for the high binding of immunoglobulins and proteins [38,39]. However, after several tests conducted on all types of plates, Nunc PolySorp microplates were chosen because of better performance.

The antigen used in the ELISA test is derived from a FCoV type I, and since it is currently impossible to discriminate FECV from FIPV viral forms, the developed test detects only anti-FCoV and not anti-FIPV antibodies in feline samples. In this context, a positive FCoV antibody test indicates that the cat has encountered FCoV (by natural infection or vaccination) and has developed antibodies. Seroconversion typically occurs around 7 to 28 days following natural infection [3].

The FCoVCHECK Ab ELISA identified samples with an excellent correlation with IFAT: the proportion of agreement was 96.4% (Cohen’s kappa of 0.925) with high sensitivity (93.5%, 95% CI: 83.5–97.9%) and high specificity (100%, 95% CI: 90.8–100%). The new ELISA was validated against IFAT as a reference method because it is considered the gold standard test for measuring anti-FCoV antibodies [11,12,13], and it was also chosen as a reference in the ImmunoComb Feline Coronavirus Antiboby kit [11].

No sample selection was performed, as no information was provided about breed, sex, age, clinical outcomes, disease progression or treatments of patients with drugs; thus, we contemplated the broadest possible cases. In this way, we also simulated the case of veterinarians who might analyze specimens from patients with unknown diagnoses or general conditions. Therefore, our ELISA assay resulted independent from breed, sex, age and physiological conditions of the animal patient. However, testing samples from cats with known experimental or field exposures against several viruses/parasites would help to reveal more about the specific mechanisms underlying cross-reactive antibody production.

The cut-off of FCoVCHECK Ab ELISA was evaluated through several methods, and it was ultimately set at OD 0.300. As reported in the instructions, samples will be classified as positive or negative by calculation of the positive and negative cut-offs based on the OD value of the negative control. The sample will be considered negative if the OD value is lower than negative cut-off, positive if it is greater than the positive cut-off and doubtful if it is between the two cut-offs. We strongly recommend instrumental reading for the correct interpretation of results since visual reading, based on the colorimetric reaction that develops after TMB addition (blue color for positive samples and no color for negative ones), is less accurate. In this context, doubtful samples will appear light blue in color, and they may be difficult to interpret (negative, doubtful or low positive) unless accompanied by a spectrophotometric reading. This type of reading should only be performed in laboratories not equipped with a microplate reader. In any case, positive and doubtful outcomes in ELISA should be carefully evaluated by clinicians and investigated or confirmed immediately by more detailed techniques, including PCR, IFAT or imaging (especially ultrasonography), taking into account the patient’s clinical history and symptoms strongly suggestive for FIP. Since antibody production time to FCoV takes anywhere from 7 to 28 days post-infection, and FCoV may preserve its infectivity for days to a few weeks, depending on environmental conditions [3], we recommend confirming the results of doubtful samples by using ELISA a few weeks after the first test.

The new ELISA is an accurate test with intra- and inter-assay %CV lower than 20%, defined as the limit of acceptable CV% for precision and accuracy in the general guidelines [25,34].

Regarding ELISA kit controls, we evaluated the use of commercial reagents according to the future production need and safety of FCoVCHECK Ab ELISA. In fact, the availability of veterinary samples, especially positives as a positive control, is limited, and the amount to be collected and put in the final kit during test production would be too much. In addition, the use of serum/plasma as a control (especially positive) might result in variability and lack of homogeneity between batches due to the nature of the sample (strong or weak positive). Moreover, to ensure operator safety, it is good practice to avoid the use of potentially infected samples, so we included in the kit only commercial reagents certified as non-infectious. In order to obtain a strong positive control, a minimum OD value of 0.6 was chosen for test validity, as reported in references [17,32] and in other ELISA assays [40,41,42]. Protein G is a protein expressed in group C and G streptococcal bacteria known to bind the constant portions of immunoglobulins [33], and because of this feature, it is widely used for immunoprecipitation assays [43,44,45] and antibody purification [46,47,48]. Protein G is already applied in some ELISA tests as a conjugate of broad specificity to detect IgG antibodies of different host species [49,50,51,52]. After preliminary assays, we demonstrated that protein G, together with the unconjugated anti-FIPV antibody, had good performance, similar to that obtained with positive samples, thus it may be a valid component of the FCoVCHECK Ag ELISA positive control. Regarding the negative control, in our developed ELISA assay, the OD values were always below the limit of 0.3, as described in references [17,32]. As negative controls, it is known that PBS-based solutions are used in both ELISA tests developed for research purposes and commercially available assays [53,54,55,56,57]. Here, we demonstrated the feasibility of using sample diluent as a negative control. In fact, the OD values of the negative control fell within the range obtained by analyzing the negative samples.

We did not observe interference in the highly lipemic or hemolytic tested samples collected in different types of anticoagulants in the FCoVCHECK Ab ELISA. We did not perform cross-reaction studies, partly due to the lack of representative samples, which are difficult to find. In this context, the presence of ELISA-interfering substances, including drugs or natural compounds in physiological or pathological conditions, should be explored. However, any cross-reactivity reactions should be carefully evaluated by clinicians based on clinical signs and symptoms of disease. We recommend that suspected false-positive and false-negative FCoV tests should be confirmed by another assay, including IFAT and PCR or imaging, considering the patient’s clinical condition. In antibody assays using indirect ELISA, false-positive and negative reactions can occur due to various reasons during sample preparation, such as errors in sample collection, storage, handling and processing and contamination, which can interfere with the accuracy of the test results. Generally, false-positive reactions may also be caused by nonspecific reactivity and background noise, resulting from the binding of immunoglobulins to the solid phase. Although we have demonstrated no aspecific binding with the coating components of the samples or controls, this reaction depends on each sample and can vary significantly, sometimes even surpassing the true antibody–antigen reaction. Although we tested a large number of samples, this step is sample-dependent and difficult to predict. In our ELISA assay, we did not detect false positives. However, considering the close genetic relationship between feline and canine coronaviruses, interspecific circulation of either CCoV in cats or FCoV in dogs is plausible. In fact, on the basis of phylogenetic analysis and antigenic cross-reactivity, the feline coronaviruses (FECV, FIPV), CCoV and TGEV display greater than 96% sequence identity within the replicase polyprotein pp1ab, and for this reason, they have been grouped in the same species within the *Alphacoronavirus* genus in the Coronaviridae family [58,59]. The FCoV I causes most infections, while FCoV II is detected only sporadically and originates from a double recombination between FCoV I and CCoV II, probably inside a co-infected cat, resulting in a genome principally composed of FCoV sequences but with the (S) gene encoding the spike (S) glycoprotein and its adjacent parts originating from CCoV [58,59,60,61]. The CCoV I spike S protein is closely related to that of FCoV I, and, similarly, that of CCoV II is close to that of FCoV II [62]. In addition, FCoV strains harboring different forms of the ORF3 gene (present only in CCoVs) and the N protein of CCoV I were identified, thus demonstrating the presence of recombinant FCoV I/CCoV I viruses [58,62]. Moreover, sequence analysis of circulating CCoVs in dogs with diarrhea in Italy revealed a new canine genetic cluster with point mutations within the M gene encoding the transmembrane protein that increased the similarity to FCoV [59,63]. In another study, cross-reactivity was observed between the S1 receptor binding subunit of the spike S protein of FCoV I and FCoV II and between FCoV I and the coronavirus porcine epidemic diarrhea virus (PEDV), thus demonstrating the presence of conserved epitopes in the domains of S1 [64]. Regarding false-negative results, we reported four false negatives in our ELISA assay compared to IFAT. Their occurrence in an antibody test is worrying, especially as FCoV antibody testing is the first method to identify FCoV infections. Negative results have been reported in three of seven cats with neurological FIP without effusions [3,65]. In other circumstances, some cats with FIP had unexpectedly low FCoV antibody titers in their effusions, with an inverse correlation between the FCoV RNA load measured by RT-PCR, while FCoV antibodies, tested with IFAT, ELISA or rapid immunochromatographic assays, were present only in some samples. One possible explanation for these false-negative results was that FCoV could bind antibodies, rendering them unavailable as a ligand in the antibody test [3,66]. Further clinical experience and additional patient outcome data will help to refine false-positive and negative results.

From the comparative study performed with two other ELISA assays widely used in the literature, the ImmunoComb test kit [11,12] and the INgezim Corona Felino ELISA [67,68,69], the FCoVCHECK Ab ELISA had the highest agreement with IFAT, considered the reference test, and more specificity than the other assays (100% specificity versus 87.5% of ImmunoComb and 64.6% of INgezim Corona Felino assays). Moreover, the new ELISA agreed more with the ImmunoComb assay (93.6%, Cohen’s kappa: 0.872, very good agreement) than with the INgezim Corona Felino ELISA (82.7%, Cohen’s kappa: 0.646, good agreement).

Controversy still exists over the interpretation of anti-FCoV antibody titers in serum or plasma since FECV and FIPV evoke the same antibody responses [70]. Cats with FIP tend to have higher FCoV antibody titers than cats without FIP, but the presence of antibodies is not useful for diagnosing FIP, since antibodies are not only present in cats with FIP but also in healthy FCoV-infected cats [8]. Many clinically healthy FECV-exposed cats and cats with FIP have titers by IFAT from 1:100 to 1:400. Healthy cats with titers <1:100 infrequently shed FECV in their feces, while cats with titers of 1:400 are usually positive for FCoV in the feces [71]. However, fewer healthy cats have titers of 1:1600, while titers ≥1:3200 are highly suggestive of FIP [72]. Despite these limitations regarding antibody titers, a titration comparative assay was performed on 21 positive samples at higher dilutions of 1:3200 up to 1:12,800, both with INgezim Corona Felino ELISA and FCoVCHECK Ab ELISA to assess the antibody titer. We obtained a good correlation with only small differences of dilution in a few samples, but in all cases, positive samples with very high antibody titers above 1:3200 suggestive of FIP were identified equally by both tests.

Although a real-world study is lacking, the company has not received any reports of test malfunction by end-users since the product was put on the market (one year).

The accelerated stability is normally performed at +37 °C to accelerate the chemical or physical reactions of the biological reagents [73,74] and it is the most suitable method for simulating product degradation and, thus, determining its expiration quickly, without waiting a long period for degradation to actually occur. In fact, many unexpected factors, such as cold-chain breakage, may occur during transportation or storage, and they may negatively impact the quality and performance of the product. Depending on country-specific regulations, stability tests may need to be conducted at defined temperatures reflecting the respective climate zone of the country in which it is intended to be sold. If the product is stable under storage conditions of a hotter climatic zone, then it is automatically suitable for use in colder zones [75,76]. A temperature of +37 °C simulates that of hotter climate zones; thus, at the end of stability testing, the product can be marketed in all climate zones, including III (hot and dry climate) and IV (hot and humid/very humid climate). The prediction of the shelf-life related to veterinary diagnostic kits (ELISA or immunochromatographic assays) does not refer to any specific legislation or regulations, unlike pharmaceutical products or veterinary drugs, for which guidelines for stability testing are present [77,78]. In the absence of standards related to veterinary diagnostic kits, reference was made to existing methods used for medical devices, as described in the Standard Guide for Accelerated Aging of Sterile Medical Device Packages ASTM F1980-21 [22], which uses the Arrhenius equation to estimate the degradation of the product under accelerated conditions. According to this, stability studies conducted on the FCoVCHECK Ab ELISA demonstrated that the kit is stable for 18 months at +2–8 °C and for 6 weeks at +37 °C, thus making the product suitable for use even under conditions of possible cold-chain breakage.

## 5. Conclusions

The FCoVCHECK Ab ELISA is an accurate test, characterized by high sensitivity and specificity. Compared to IFAT as a reference, the new assay correctly identifies positive and negative samples with a good correlation and, in addition, it is simpler, faster and provides a less subjective reading of the results. The new ELISA provides correct and reliable results quickly (1 h), without the need for thermostats. The interpretation of results is performed instrumentally by reading the OD values at 450 nm using an ELISA microplate reader. The new test is also suitable for the analysis of highly lipemic or hemolytic samples, as no interference was observed in this study. The kit contains safe reagents for the end-user and has a long-term storage up to 18 months at +2–8 °C. Diagnosis of FIP is very complex, but given the good correlation with IFAT, the new assay may be a valuable aid to clinicians in evaluating FCoV infections and in including or excluding the possibility of FIP in veterinary practice.

## Figures and Tables

**Figure 1 vetsci-12-00245-f001:**
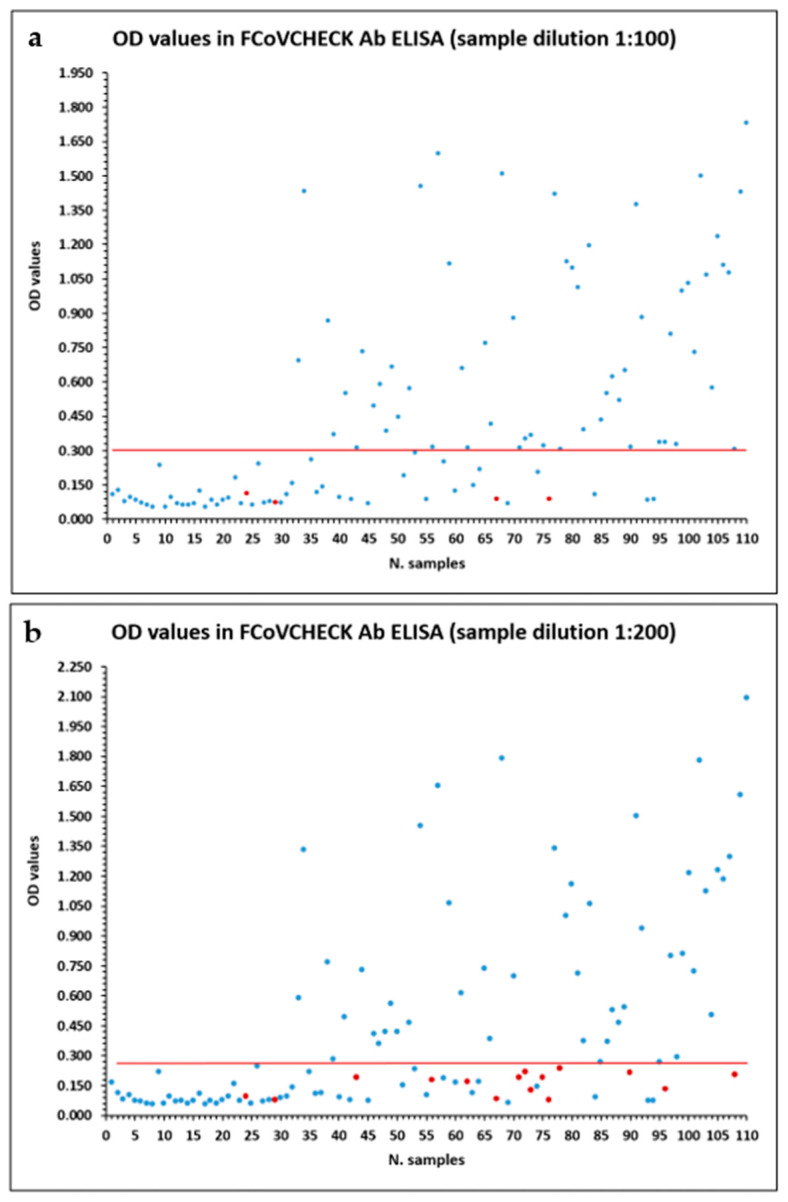
Overall graphs of OD values obtained by sample dilution 1:100 (**a**) or 1:200 (**b**). Red dots indicate the discordant false-negative data in ELISA compared with the reference IFAT method. The red line corresponds to the OD cut-off level that best separates positive and negative samples.

**Figure 2 vetsci-12-00245-f002:**
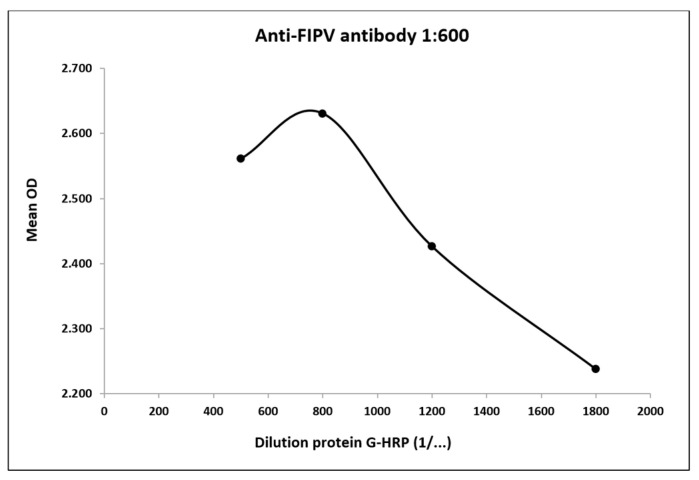
Positive control selection. Performance of the anti-FIPV antibody tested at 1:600 and protein G-HRP at dilutions ranging from 1:500 to 1:1800.

**Figure 3 vetsci-12-00245-f003:**
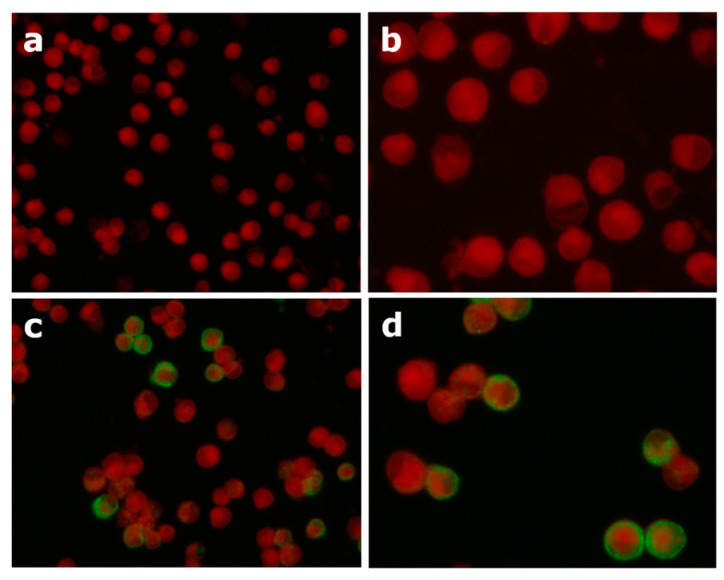
Example of negative (**a**,**b**) and positive (**c**,**d**) test results in IFAT observed at 100X (**a**,**c**) and 400X magnification (**b**,**d**).

**Figure 4 vetsci-12-00245-f004:**
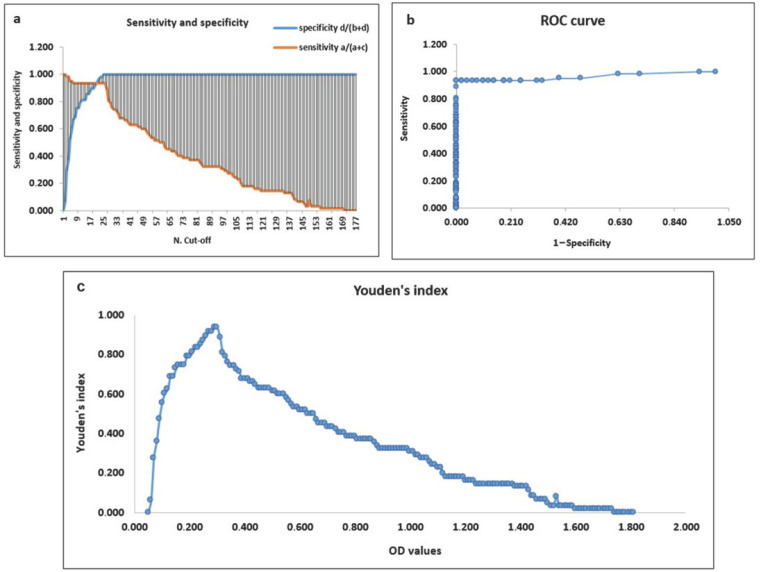
Methods for cut−off determination. Cut−off was determined by evaluating (**a**) sensitivity and specificity parameters, (**b**) ROC curve and (**c**) Youden’s index.

**Figure 5 vetsci-12-00245-f005:**
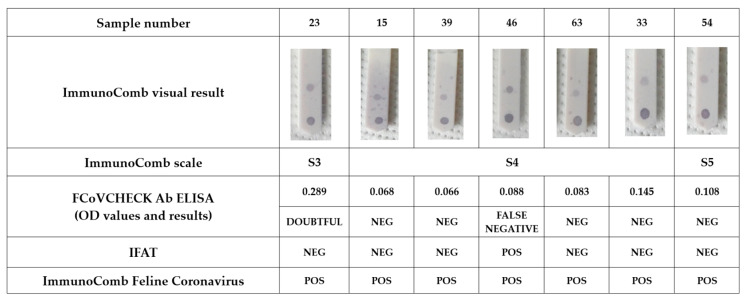
Comparison results on false-positive samples in ImmunoComb assay (Biogal). In ImmunoComb, the upper spot corresponds to the positive control, the lower spot to the sample. CombScale scores: S3 and S4 (positive, FIP possible); ≥S5 (strong positive, high probability of FIP). Results of FCoVCHECK Ab ELISA are indicated as optical density. POS: positive; NEG: negative.

**Table 1 vetsci-12-00245-t001:** Absorbance values obtained on Nunc PolySorp microplates coated with 0.5, 1 and 2 µg/mL of antigen and conjugate antibody at dilutions ranging from 1:2500 to 1:10,000. Samples: one positive, one negative (dilution 1:200, 50 µL/well); n.d.: not determined.

Conjugate Dilution	Positive Sample			Negative Sample
	0.5 µg/mL	1 µg/mL	2 µg/mL	2 µg/mL
1:10,000	0.178	0.322	0.660	n.d
1:5000	0.272	0.409	1.134	n.d
1:2500	0.489	0.911	1.849	0.056

**Table 2 vetsci-12-00245-t002:** Test ELISA on Nunc PolySorp microplates coated with 1 µg/mL of antigen and conjugate antibody 1:2500 on nine samples (5 negatives, 4 positives) at dilution 1:200 (100 µL/well). PC: positive control (positive serum, 1:200); NC: negative control.

Samples	OD Values
Negative 1	0.079
Negative 2	0.083
Negative 3	0.076
Negative 4	0.122
Negative 5	0.070
Positive 1	1.073
Positive 2	1.051
Positive 3	0.973
Positive 4	1.081
PC	1.419
NC	0.052

**Table 3 vetsci-12-00245-t003:** Sample dilution selection in relation to IFAT as a reference test. A total of 110 samples were analyzed at dilutions 1:100 or 1:200. (a) True positives; (b) false positives; (c) false negatives; (d) true negatives.

Sample Dilution	FCoVCHECK Ab ELISA	IFAT		
		Positive	Negative	Total
1:100	Positive	58 (a)	0 (b)	58
	Negative	4 (c)	48 (d)	52
	Total	62	48	110
1:200	Positive	47 (a)	0 (b)	47
	Negative	15 (c)	48 (d)	63
	Total	62	48	110

**Table 4 vetsci-12-00245-t004:** Aspecific binding and functionality of the positive control. PC: anti-FIPV antibody (1:50–1:10,000); conjugate: anti-Cat IgG HRP antibody 1:3000 or anti-Goat IgG HRP antibody 1:2000.

Samples	Sample Dilution	Conjugate Antibody	
		Anti-Cat HRP	Anti-Goat HRP
Negative sample	1:200	0.047	0.049
Positive sample	1:200	2.609	0.051
PC	1:50	0.045	3.509
	1:150	0.048	3.259
	1:500	0.051	2.202
	1:1500	0.048	2.752
	1:5000	0.046	1.743
	1:10,000	0.053	1.055

**Table 5 vetsci-12-00245-t005:** Summary of the reproducibility study. For each sample, the overall number of tests (N); mean, median, minimum (Min), maximum (Max) and mode OD values; variance; standard deviation (SD); and %CV, including that for the intra- and inter-assay, were reported. PC: positive control; NC: negative control; samples: 1, 2, 3 (negatives) and 4 (positive).

Samples	N.	Mean	SD	%CV	Min	Max	Median	Variance	Mode	Intra-Assay %CV		Inter-Assay %CV	
										Min	Max	Min	Max
PC	56	2.662	0.014	0.518	2.626	2.689	2.666	0.000186	2.654	0.053	0.610	0.079	0.426
NC	24	0.043	0.002	4.063	0.041	0.049	0.043	0.000003	0.043	0.000	7.603	0.832	3.802
Sample 1	24	0.057	0.003	5.002	0.052	0.061	0.056	0.000008	0.055	0.000	4.962	0.626	2.481
Sample 2	24	0.053	0.002	3.793	0.049	0.057	0.053	0.000004	0.052	0.000	7.857	0.000	5.839
Sample 3	24	0.076	0.006	7.379	0.072	0.094	0.074	0.000030	0.073	0.000	11.279	0.488	6.595
Sample 4	24	0.710	0.071	9.977	0.567	0.851	0.691	0.004814	-	0.105	19.373	1.402	12.523

**Table 6 vetsci-12-00245-t006:** Comparison of FCoVCHECK Ab ELISA (Agrolabo) with INgezim Corona Felino ELISA (Gold Standard Diagnostics, GSD) and ImmunoComb (Biogal). Number of samples tested: 110; (a) true positives; (b) false positives; (c) false negatives; (d) true negatives.

Assay		FCoVCHECK Ab ELISA (Agrolabo)		
		Positive	Negative	Total
ImmunoComb (Biogal)	Positive	58 (a)	7 (b)	65
	Negative	0 (c)	45 (d)	45
	Total	58	52	110
INgezim Corona Felino ELISA (GSD)	Positive	58 (a)	19 (b)	77
	Negative	0 (c)	33 (d)	33
	Total	58	52	110

**Table 7 vetsci-12-00245-t007:** Comparative study and statistical analysis. Statistical difference was assessed by *t*-test (*p*-value ≤ 0.05) on FCoVCHECK Ab ELISA (Agrolabo S.p.A) and INgezim Corona Felino ELISA (Gold Standard Diagnostics, GSD). OD: optical density. Cohen’s d (Effect size): d = 0.2 small effect, d = 0.5 moderate effect, d = 0.8 great effect.

	Positive Samples		Negative Samples	
	ELISA (Agrolabo)	INgezim Corona Felino (GSD)	ELISA (Agrolabo)	INgezim Corona Felino (GSD)
N. Tests	58/58	58/58	52/52	33/52
Mean OD	0.775	0.787	0.112	0.139
*p*-value (≤0.05)	0.879		0.052	
Cohen’s d	-		0.437	

**Table 8 vetsci-12-00245-t008:** Comparison results of FCoVCHECK Ab ELISA, INgezim Corona Felino ELISA and ImmunoComb against IFAT. Total number of samples analyzed: 110; (a) true positives; (b) false positives; (c) false negatives; (d) true negatives.

Samples	IFAT	FCoVCHECK Ab ELISA (Agrolabo)	ImmunoComb (Biogal)	INgezim Corona Felino ELISA (Gold Standard Diagnostics)
a	62	58	60	60
b	0	0	6	17
c	0	4	2	2
d	48	48	42	31
Sensitivity		93.5% (CI 95%: 83.5–97.9%)	96.8% (CI 95%: 87.8–99.4%)	96.7% (CI 95%: 87.8–99.4%)
Specificity		100% (CI 95%: 90.8–100%)	87.5% (CI 95%: 74.1–94.8%)	64.6% (CI 95%: 49.4–77.4%)
Cohen’s kappa		0.925	0.851	0.636

**Table 9 vetsci-12-00245-t009:** Accelerated stability study. Analysis times: T_0_–T_6_. OD: mean of spectrophotometric readings; %RA: percentage remaining activity; PC: positive control; NC: negative control. Samples 1, 2, 3: negatives; Sample 4: positive.

Samples	T_0_		T_1_		T_2_		T_3_		T_4_		T_5_		T_6_	
	OD	%RA	OD	%RA	OD	%RA	OD	%RA	OD	%RA	OD	%RA	OD	%RA
PC	2.867	100	2.745	95.76	2.625	91.58	2.631	91.77	2.662	92.87	2.606	90.91	2.552	89.01
NC	0.043	100	0.047	108.14	0.048	111.63	0.048	110.47	0.048	110.47	0.047	109.30	0.045	103.49
Sample 1	0.048	100	0.049	102.08	0.051	106.25	0.050	103.13	0.053	110.42	0.048	100.00	0.049	101.04
Sample 2	0.057	100	0.059	103.54	0.063	111.50	0.059	104.42	0.061	107.08	0.058	101.77	0.056	98.23
Sample 3	0.174	100	0.172	98.56	0.173	99.14	0.150	85.92	0.157	90.23	0.137	78.45	0.148	85.06
Sample 4	0.520	100	0.539	103.66	0.550	105.87	0.460	88.55	0.462	88.84	0.432	83.06	0.444	85.37

## Data Availability

Data are available in this published article and as Appendix A. Datasets generated during the current study are also available from the corresponding author on reasonable request.

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
