# Peer review of "Development of a New Indirect ELISA Test for the Detection of Anti-Feline Coronavirus Antibodies in Cats"

_vetsci, 2025, doi:10.3390/vetsci12030245_

Round 1

Reviewer 1 Report

Comments and Suggestions for Authors

Thank you for the opportunity to review this manuscript.

This article aimed to develop an ELISA test for the detection of antibodies anti-FCoV as a diagnostic support for FIP in feline serum and plasma samples. An assay was meticulously established with great sensitivity and specificity and seems to perform even better than the two other commercial rapid ELISA tests to which it was compared. Due to its rather fast, less labor intensive and less subjective way, it might provide a valuable alternative to IFAT and thus an aid in evaluating FCoV antibody titers in cats with FIP suspicion. This study is interesting and well-done. However, there are several points which might improve this manuscript.

Please find my comments and suggestions in the attached file.

Author Response

Thank you very much for your feedback and your time in reviewing this manuscript.

Below are the detailed point-to-point responses to comments and suggestions, highlighted in yellow in the manuscript.

Title

The title of the manuscript was changed by removing the recommended part. Thus, the title will be “Development of a new indirect ELISA test for the detection of anti-feline coronavirus antibodies in Cats”.

  • The postal codes were added to all affiliations
  • The Simple Summary has been added before Abstract.

Abstract

  • Line 25-28: the sentence regarding FCoV pathotypes has been revised as recommended.
  • Line 39-31: the sentence regarding IFAT as gold standard method has been revised.
  • Line 33: your comment regarding visual and instrumental readings is correct.

The instrumental reading is strongly recommended for the correct interpretation of results. In this manuscript we did not investigate the part about the visual reading because it is less accurate than spectrophotometric reading and can only be performed in laboratories not equipped with a microplate reader. However, we still evaluated it internally at each analytical session. A visual interpretation is based on the colorimetric reaction that develops after TMB addition as described in Section 2.3 "FCoVCHECK Ab ELISA": a blue colour in the well is index of positive samples that contain anti-FCoV antibodies, no colour of negative ones. In this context, doubtful samples will appear in light-blue colour and they may be difficult to interpret unless accompanied by an instrumental reading.  

To avoid confusion of the reader and, therefore, to the potential end user of the kit, we prefer to remove the sentence “with both visual and spectrophotometric reading” in the Abstract and insert the comment in the Discussion (line 602 – 608).

  • Line 41-43: we agreed with your comment. The sentence has been changed.

  1. Introduction
  • Line 58-62: the sentence has been added as recommended. “The widely accepted hypothesis is that FECV converts to FIPV, so genetic variation and subsequent selection facilitate the switching of cell tropism from enterocytes to systemic monocytes/macrophages within an FCoV-infected cat. Effective replication of the virus within this type of leukocytes is necessary for the development of FIP
  • Line 62: the sentence has been revised, the reference number 5 has been changed by including the work of Porter et al. (2014) and added in the References.
  • Line 68: we prefer “titers”
  • Line 73 – 77: the sentence regarding FIP forms has been revised.
  • Line 80: the reference for T cell depletion has been added.
  • Line 83: the sentence along with thoracic and pericardial effusion” has been added.
  • Line 88 – 91: the sentence regarding the measurement of anti-FCoV antibodies has been revised.
  • Line 92 – 94: the reference number 9 has been added.
  • Line 100 – 104: the sentence has been revised.

  1. Materials and Methods
  • Line 107 – 114: the sentence about samples has been added.
  • Line 118: the reference about IFAT method was added.
  • Line 119: Lipemia of samples was evaluated only visually as the turbidity of the sample caused by accumulation of lipoprotein particles in serum and plasma samples. In fact, visual detection of lipemia in the patient samples is a widely used approach, especially in the laboratories with low number of samples (Nikolac, 2014).

As reported in line 635 – 636 (Discussion) and line 742 – 743 (Conclusions), we did not perform extensive studies such as spike-in experiments to verify cross-reactivity reactions. We did not observed interference on the highly lipemic or haemolytic samples that we tested, but we do not rule out the absence of interference on other samples. This element would be verified in future studies.

Reference for lipemia:

- Nikolac N. Lipemia: causes, interference mechanisms, detection and management. Biochem Med (Zagreb). 2014, 24, 57-67. https://doi.org/10.11613/bm.2014.008

  • Line 129: the information about the FCoV Type I has been added.
  • Line 133 – 135: the name of the Blocking buffers has been corrected, making it like the Wash buffer.
  • Line 153: the information about the anti-Cat IgG conjugate antibody has been added.
  • Line 161: the information about ELISA microplate reader has been added.
  • Line 163: the sentence about the cut-off has been added.
  • Line 172: IFAT was considered the reference test for serologic examination. In fact, the gold standard in FCoV antibody tests is generally regarded as the IFAT (Addie et al., 2015). I added this reference in the text.
  • Line 180: the information about fluorescence microscope has been added.
  • Line 187 - 190: the ELISA commercial names were already included.
  • Line 191: the sentence “according to the manufacturer’s protocol” has been added.

  1. Results
  • Table 1 and S1: The description has been updated.
  • Table 5, Table 9 and Table 12 were put in the Supplementary files as recommended. Regarding Table 12, all its information was inserted in Supplementary Table S10. I revised all number of Tables in the manuscript.
  • Supplementary Table S3: good results were selected considering the costs and performances reasons in terms of OD values (OD > 0.6 and not over-flowing). I added this explanation in the description.
  • Figure 1b: sorry, the red dot it was a mistake. That point doesn't exist. It must have been an error that occurred during the marking of the other points. Thanks for the observation. The false-negative results are 15 points, already highlighted below the red line in the figure.
  • Figure 5: I modified the order of the rows as recommended.
  • Supplementary Table S8: I added the cut-offs values of both assays in the description.

  1. Discussion
  • Line 563 - 564: a short introduction sentence has been added to start the Discussion.
  • Line 577: the part regarding the “FCoV-infected cases” has been reformulated.
  • Line 580: I added the reference for seroconversion that occurs from 7 to 28 days following natural infection.
  • Line 618: I agree with you: one of the preparations that can be implemented is heat inactivation of sera. However, the major limitation, among those reported in the manuscript to the use of sera as positive controls, is the amount to be collected and put in the final kit during test production that would be too much and not sustainable for production needs.
  • Line 674 – 676: since we developed a qualitative ELISA assay, we did not perform a titration assay in IFAT. In vast, during IFAT, we tested only the first sample dilution 1:100 according to the manufacturer’s instructions and we assessed if samples were positive or negative. Then, we validated the test by verifying the correspondence of the result obtained for each sample in ELISA with those achieved by IFA. Thus, we cannot affirm that the false-negatives obtained in ELISA were related to low titers in IFAT.
  • Line 698 – 699: It is correct. Both clinically healthy FECV-exposed cats and cats with FIP have titers by IFAT from 1:100 to 1:400. I reformulated the sentence.

  1. Conclusion
  •  Line 737 – 739: I added the sentence regarding the advantages of ELISA vs IFAT.

Reviewer 2 Report

Comments and Suggestions for Authors

The purpose of this study FCoVCHECK Ab ELISA, a new ELISA test for the detection of antibody anti-feline coronavirus as diagnostic support for Feline Infectious Peritonitis (FIP). This study is important to diagnostic of FIP is very complex and requires various investigations since existing diagnostic tests cannot differentiate between feline enteric coronavirus (FECV) and feline infectious peritonitis virus (FIPV).

This study demonstrates that FCoVCHECK Ab ELISA is an accurate test, characterized by high sensitivity and specificity. 

Author Response

The authorship has not been changed